# Social Reward: Evaluating and Enhancing Generative AI through Million-User Feedback from an Online Creative Community

**Arman Isajanyan**[1][*] **Artur Shatveryan**[1][*] **David Kocharyan**[1], **Zhangyang Wang**[1,2], **Humphrey Shi**[1,3]
[1]Picsart AI Research (PAIR), [2]UT Austin, [3]Georgia Tech
{arman.isajanyan, artur.shatveryan, david.kocharyan,
 atlas.wang, humphrey.shi}@picsart.com

## Abstract

Social reward as a form of community recognition provides a strong source of motivation for users of online platforms to actively engage and contribute with content to accumulate peers approval. In the realm of text-conditioned image synthesis, the recent surge in progress has ushered in a collaborative era where users and AI systems coalesce to refine visual creations. This co-creative process in the landscape of online social networks empowers users to craft original visual artworks seeking for community validation. Nevertheless, assessing these models in the context of collective community preference introduces distinct challenges. Existing evaluation methods predominantly center on limited size user studies guided by image quality and alignment with prompts. This work pioneers a paradigm shift, unveiling **Social Reward** - an innovative reward modeling framework that leverages implicit feedback from social network users engaged in creative editing of generated images. We embark on an extensive journey of dataset curation and refinement, drawing from *Picsart*: an online visual creation and editing platform, yielding **a first million-user-scale** dataset of implicit human preferences for user-generated visual art named **Picsart Image-Social**. Our analysis exposes the shortcomings of current metrics in modeling community creative preference of text-to-image models' outputs, compelling us to introduce a novel predictive model explicitly tailored to address these limitations. Rigorous quantitative experiments and user study show that our Social Reward model aligns better with social popularity than existing metrics. Furthermore, we utilize Social Reward to fine-tune text-to-image models, yielding images that are more favored by not only Social Reward, but also other established metrics. These findings highlight the relevance and effectiveness of Social Reward in assessing community appreciation for AI-generated artworks, establishing a closer alignment with users' creative goals: creating popular visual art. Codes can be accessed at https://github.com/Picsart-AI-Research/Social-Reward

## 1 Introduction

Social reward mechanisms play a pivotal role in incentivizing and modulating human behavior. Grounded in neurobiology and psychology, positive social feedback, such as approval, validation, and recognition, are essential for maintaining social cohesion and individual well-being (Rudolph, 2021; Baumeister & Leary, 1995). This reward-driven behavior extends to online social platforms, where users seek satisfaction via the accumulation of their network's peer engagement with shared content in forms such as likes or views (Deters & Mehl, 2013; Lemai Nguyen & Nallaperuma, 2023).

Recently, the field of text-conditioned image synthesis has witnessed remarkable advancements, leading to the development of generative algorithms capable of producing high-fidelity images that closely adhere to textual descriptions. This technological breakthrough has significantly impacted the realm of online social networks, as it empowers users with a novel and creative means of content creation and sharing. As users leverage this technology to craft and post compelling visual content,

---

[*]Equal contribution

| Prompt | Social Reward | HPS v2 | Image Reward | PickScore |
|---|---|---|---|---|
| *Digital anime art of mattress-man with a serious expression in an empty warehouse, highly detailed* | | | | |
| *energizing morning routines* | | | | |
| *princess crown* | | | | |
| *A beautiful woman standing in a dystopian city* | | | | |

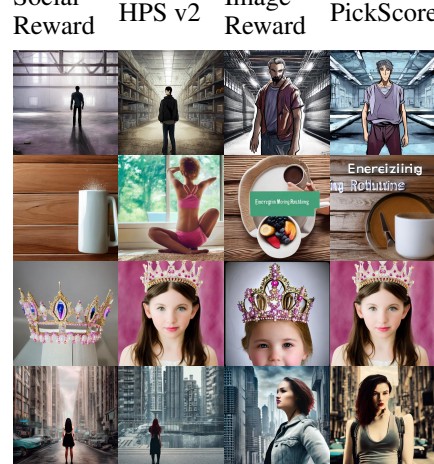

Figure 1: Best image out of 20 generations as chosen by different scoring models, including ours.

they simultaneously tap into the well-established reward mechanisms of social validation and recognition. Given the pace with which the number of synthetic images grows filling the digital spaces of online creative communities (as of August 2023 more than 15 billion synthetic images had been generated globally (Valyaeva, 2023)) evaluating the performance of generative models within the context of social network popularity emerges as an important challenge.

While social network popularity can be defined in many ways, with likes, views, and other similar types of user interactions traditionally serving as popularity estimates (McParlane et al., 2014; Ding et al., 2019), the nature of text-to-image technology, adopted by industry largely as co-editing tool integrated into creative platforms (Weisz et al., 2023; Huang & Grady, 2022), introduces another dimension to social network content popularity measurement, namely the frequency of synthetic image reuses for editing purposes by community members. This metric resonates with the population of artists and creators receiving social rewards when their synthetic images are being leveraged in the editing process by network peers. The central question then can be summarized as, *to what extent text-to-image models can produce visual content aligned with social popularity, which is defined as community preference for creative editing purposes?*

Recently, researchers have increasingly turned to human preferences as a guiding beacon, inspired by the transformative impact of human feedback in the realm of Large Language Models (LLM) (Ouyang et al., 2022; Nakano et al., 2022). In the domain of text-to-image generation, reward models have been harnessed to channel human feedback into the learning process Xu et al. (2023a); Wu et al. (2023a); Kirstain et al. (2023). These works, have sought to leverage human preferences to construct reward models that facilitate generative model evaluation and fine-tuning.

Despite the commendable efforts, the existing reward models have notable limitations in our domain of interest. Some of them rely on limited-size data annotation process, as presented in Table 1, which cannot be deemed the equivalent of the "community-scale" feedback. Moreover, prompts utilized for dataset creation (collected from COCO Captions dataset (Chen et al., 2015) and open source prompt dataset DiffusionDB (Wang et al., 2023)) along with the moderation process and guidelines that adhere mainly to image fidelity and textual alignment as annotation criteria, do not emphasize creative purpose and hence, potentially, are limiting in expressing collective community creative preference. On the other hand, some other approaches collect explicit organic user feedback, but as a downside, exhibit a relatively small scale of collected user preference and absence of "collective feedback" (when more than one user engages with a given image) as an important indicator of social popularity. While these approaches do capture a broad spectrum of user preferences, they nevertheless showcase insufficiency to model social popularity in the context of community-level editing preference. These limitations are substantiated by our extensive quantitative and qualitative analysis.

To bridge this gap and address the unique demands of text-to-image in the framework of creative social popularity, we introduce a novel concept: **Social Reward**. This **paradigm shift in reward modeling** leverages collective implicit feedback from social network users who specifically employ generated images for creative purposes. This distinction underscores the relevance and applicability

of Social Reward to the creative editing process, providing a more faithful estimation of alignment between AI-generated images and community-level creative preference. However, the collection of Social Reward data presents its own set of challenges, notably the inherent noise stemming from the implicit nature of user feedback, the absence of a formal annotation process with precise guidelines, and the unequal content exposure caused by social network-specific factors (such as some content being surfaced more frequently than the others, etc).

In this work we embark on a comprehensive exploration, starting with data curation sourced from **Picsart** (`https://picsart.com/`): one of the world's leading online visual creation and editing platforms. Due to the inherent noise and subjectivity in individual user choices, the collective feedback, which implies multiple users' editing choices for the given content item, is leveraged as a cleaning mechanism of organic implicit user behavior. Several more data collection techniques have been utilized for addressing such biases as caption bias, content exposure time, and user follower base biases. Our analysis reveals the shortcomings of existing metrics in evaluating text-to-image models' fitness for generating popular art, which motivates us to introduce a new model explicitly designed to address these limitations. Moreover, we demonstrate the potential of our model in fine-tuning text-to-image models to better align with community-level creative preference.

Our contributions are outlined as follows:

- We identify an unexplored, but extremely relevant dimension in human preference reward modeling for text-to-image models: evaluating the performance within the context of social network popularity for creative editing. Our analysis provides compelling evidence that existing reward models are ill-suited to capture this dimension.

- We embark on a journey of dataset curation and leveraging *Picsart's* creative community data. We build a large scale dataset of implicit human preferences motivated by creative editing intent over synthetic images, named **Picsart Image-Social** dataset. Contrary to existing methods, we utilize social network user feedback and curate dataset relying on editing community collective behavior.

- Building upon this curated dataset, we develop and validate our **Social Reward Model**, showcasing its superiority for the given task, as evidenced in Table 4 and Figure 5. Furthermore, our model captures distinct image attributes that go beyond mere aesthetics (see Figure 1), demonstrating its potential to enhance text-to-image model performance for community-level creative preference (see Table 5 and Figure 7).

## 2 RELATED WORK

### 2.1 TEXT-TO-IMAGE GENERATION

Text-to-image generative models allow synthesizing images conditioned on the text input. GANs (Goodfellow et al., 2014) allowed for the first successful results in this realm (Zhang et al., 2017; Xu et al., 2017; Zhu et al., 2019; Liao et al., 2022). Transformer-based (Ramesh et al., 2021; Chang et al., 2023) models have also yielded great improvements. Recently diffusion-based architectures showed great ability of producing high fidelity images (Nichol et al., 2022; Ramesh et al., 2022; Xu et al., 2023b; Lu et al., 2023). LDM (Rombach et al., 2022) employs a diffusion process in the underlying latent space rather than directly in the pixel space. This approach delivers notable performance gains while also enhancing processing speed.

### 2.2 POPULARITY PREDICTION

The domain of predicting content popularity within social networks has garnered substantial attention in recent years, primarily due to its relevance in comprehending content diffusion dynamics, modeling community preference patterns, and the optimization of marketing strategies. Most existing works focus on the following types of media content: text (Oza & Naik, 2016; Gao et al., 2019), video (Li et al., 2013; Rizoiu et al., 2017) and images (Khosla et al., 2014)]. In the realm of image popularity prediction most existing works use either Flickr (McParlane et al., 2014) or Instagram (Ding et al., 2019) for their dataset curation. While significant research has been done on social content popularity topic, little attention has been put on the emerging field of synthetic/generated images and their fitness for popularity in a creative community.

## 2.3 TEXT-TO-IMAGE EVALUATION

Given that the focus of this study lies in the realm of evaluating the performance of text-conditioned image generation models, it is imperative to assess the latest advancements in related research. While widely accepted evaluation methods within the image synthesis community, commonly referred to as "automated metrics" like FID (Heusel et al., 2018) and CLIP score (Radford et al., 2021), have demonstrated certain drawbacks (Otani et al., 2023; Parmar et al., 2022; Xu et al., 2023a; Kirstain et al., 2023), one notable concern is their limited alignment with human preferences.

To tackle this issue, recent endeavors (Kirstain et al., 2023; Xu et al., 2023a; Wu et al., 2023a) have proposed direct modeling of human preference by learning scoring functions from datasets consisting of human-annotated prompts paired with synthetic images. While these studies represent a significant stride towards enabling practitioners to approximate human preference for generative model outputs, they still exhibit several inherent limitations concerning the core objective of our research, which revolves around addressing the challenge of predicting social popularity.

- A subset of the prompts used in HPD v2 (Wu et al., 2023a) is sourced from the COCO Captions dataset, which comprises captions linked to real images. This incorporation raises concerns about a potential domain mismatch when evaluating the scoring of generated images. Furthermore, the gathered feedback stems from a relatively **restricted** number of annotators, rendering it insufficient to encapsulate the preferences of a large-scale user base. In contrast, our feedback is drawn from a user community numbering in the **millions** individuals who actively engage with these images for editing purposes.

- Likewise, ImageReward (Xu et al., 2023a) dataset faces limitations not only in terms of annotators number but also in terms of a relatively low number of unique prompts and images.

- One important shared concern by both ImageReward and HPD v2 is the absence of specific guidance to direct annotators toward emphasizing creative editing goals. Instead, their focus was primarily on ensuring image fidelity and aligning text with images.

- We performed a prompt analysis to compare prompts derived from the datasets used for training scoring models, which includes prompts from ImageReward and Pick-a-Pic (Kirstain et al., 2023)[1], with prompts crafted by our platform's creators, which by our popularity metric are reflective of creative image editing intent. It is evident that prompts in ImageReward and Pick-a-Pic datasets significantly deviate from those leveraged for creative editing.

- In Pick-a-Pic case prompts are generated by web app, created by paper authors for research purposes. Users had been invited to interact with applications via such channels as Twitter, Facebook, Discord, and Reddit. The relatively small scale of collected user preferences along with the absence of "collective feedback" (understood as different users' independent choice to interact with a given image) make Pick-a-Pic less optimal approach for community popularity modeling.

Table 1: Comparison of human preference datasets for text-to-image evaluation

| Name | Annotator type | Annotator focus | Prompt source | Image source | Number of Images | Image Pairs | Distinct Prompts | Users/ Annotators |
|---|---|---|---|---|---|---|---|---|
| HPD v2 | Professional annotators | Image Quality + Text Alignment | COCO captions + DiffusionDB | 9 T2I models + COCO Captions | 430K | 798K | **104K** | 57[*] |
| ImageReward | Professional annotators | Image Quality+ Text Alignment | DiffusionDB | Stable Diffusion | 55K | 137K | 8.9K | 24[**] |
| Pick-a-Pic | Real users | Individual Feedback | Pick-a-Pic Web app | Different models/configs | 656K | 615K | 38.5K | 6.4K |
| Picsart Image-Social | Real users | Social Feedback | Social platform user prompts | Several inhouse models | **1.7M** | **3M** | **104K** | **1.5M** |

[*] 7 of them are quality checkers.

[**] After annotation quality inspectors double-checked each annotation, and those invalid ones were assigned to other annotators for relabeling.

---

[1]HPD v2 by the time of this paper writing didn't share the training part of the dataset

## 3  SOCIAL PREFERENCE: NEW DATASET CURATION AND ANALYSIS

### 3.1  PICSART AS EDITING-CENTRIC CREATIVE COMMUNITY

*Picsart* stands as one of the world's largest digital creation platforms, encompassing a wide spectrum of AI-driven editing tools, notably including text-to-image capabilities. This comprehensive suite empowers creators of varying proficiency levels to conceive, refine, and disseminate photographic and video content. Central to the platform's appeal is its robust open-source content ecosystem, perpetually invigorated by a vibrant user community.

*Picsart* serves both personal and professional design needs, distinguished by a distinct social dimension. This social facet allows users to share their creative edits, which, in turn, can be harnessed by fellow members of the platform. Consequently, the popularity of a user escalates when they share images that find utility among their peers. *Picsart* also incorporates a sophisticated search component, enabling users to locate and utilize one another's edits effectively.

### 3.2  DATA COLLECTION

In the realm of popularity prediction, most studies commonly employ metrics such as comments, views, or likes as labels for prediction (McParlane et al., 2014; Khosla et al., 2014; Ding et al., 2019). However, *Picsart's* distinct creative nature has prompted us to explore a rather unique metric, called *remix*: **the number of times an image has been reused for editing purposes by other users**. This intriguing metric represents one of the most prevalent editing behaviors within our community. What sets remixing apart from conventional popularity signals is its deeper level of user engagement. It is not merely a passive indicator, but rather a testament to active involvement, that allows discerning, which synthetic images hold greater appeal for transformative and artistic modifications. Essentially, our community implicitly conducts a form of collective voting, determining the fitness of prompt-image pairs for creative editing.

We have established specific criteria for identifying positive (popular) and negative (unpopular) images associated with a given prompt, grounded in the following community-driven editing signals:

- **Content Signal**: We consider the frequency with which a given image has been remixed by members of the community.
- **Creator Signal**: When an image is remixed by top influencer artists within our community.

In addition to visual and textual attributes, there exists a multitude of factors influencing image popularity. Much like the approach followed by Ding et al. (2019), we have taken careful measures in collecting data to mitigate potential sources of biases:

- **Prompt Bias**: To reduce the impact of the prompt that can cause non-even content distribution in the platform, we condition our model on the prompt, accompanied by a pair of popular and non-popular images.
- **Content Exposure Time Bias**: Since some images receive higher viewership than others, for the given prompt we select unpopular images with relatively close exposure time to popular ones.
- **User Follower Base Bias**: Difference in the size and engagement of individual user follower bases does not affect the popularity of generated images, because all those images are posted under the same *Picsart* public profile.

Our meticulously collected dataset[2], known as the **Picsart Image-Social dataset** is represented by triplets: prompt, positive image, and negative image. Additionally, mature images were filtered out by in-house NSFW detection algorithms. For a more detailed description of the data collection procedure, please refer to Appendix A.2. With careful consideration given to avoiding prompt-level intersections, the dataset is partitioned into training (70%), validation (10%), and test (20%) sets. High level statistics of **Picsart Image-Social dataset** can be found in Table 1. For additional information please refer to Appendix A.3.

---

[2]Picsart is a company that complies with GDPR, CCPA, and other data protection legislation and is collecting, storing, and processing users' data by the consent received. It also allows the users to opt out of certain processing purposes, as requested.

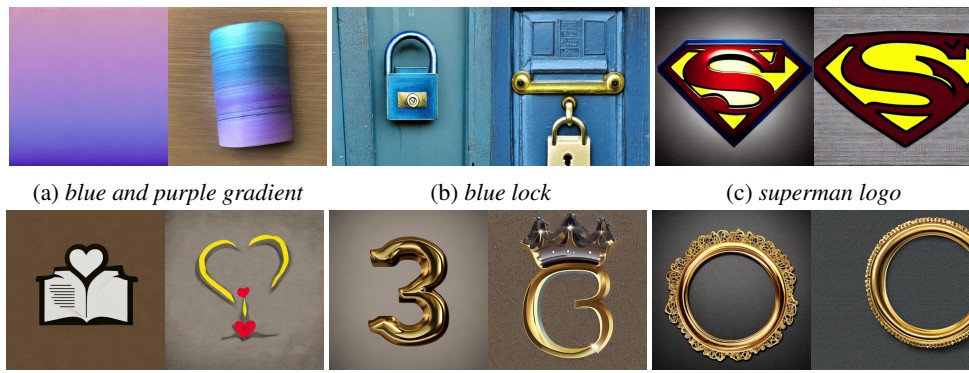

(a) *blue and purple gradient*  (b) *blue lock*  (c) *superman logo*

(d) *a logo combining a book with a heart*  (e) *the number 3 in shiny gold liquid thick font...*  (f) *golden round frame on black background*

Figure 2: Picsart Image-Social training set, showing popular (*left*) and non-popular images (*right*)

Leveraging our social network community has allowed us to amass a larger dataset than parallel text-to-image human preference modeling efforts. Despite the implicit nature of human preferences in our dataset (users edit images organically for creative purposes) the substantial feedback volume minimizes the risk of noisy signals. Each instance in the Picsart Image-Social dataset represents collective, independent, implicit voting by our user community, rather than the preference of a single member. Examples of our collected data are illustrated in Figure 2.

## 3.3 Empirical analysis of existing evaluation frameworks

To demonstrate the distinctiveness of the Picsart Image-Social dataset in comparison to those employed by existing solutions and to highlight the limitations of these solutions, we undertook a comprehensive analysis. Our three-fold analysis (prompt analysis, comparative quantitative evaluation on collected test set and comparative label explainability analysis of score models datasets) reveals that current evaluation models are poorly suitable for estimating the social popularity of images.

### 3.3.1 Prompt Analysis

Training datasets of Picsart Image-Social, ImageReward and Pick-a-Pic have different sources which also indicates distinct motives for image generation. To illustrate Picsart Image-Social's uniqueness and that it reflects creative editing intent, we conducted a cluster analysis on the Sentence-BERT (Reimers & Gurevych, 2019) embedding of the prompts in the training datasets. For fair comparison we sampled an equal amount (6000 unique prompts) of data from each dataset, filtered non-english prompts and truncated prompts to only include the first 5 words. Clustering was done using Ward's hierarchical clustering method (Ward, 1963), identifying 173 clusters.

Table 2: KL divergence of training prompts for different scoring models

| Model 1 | Model 2 | KL |
|---|---|---|
| Picsart Image-Social | Pick-a-Pic | 0.6 |
| Picsart Image-Social | ImageReward | 0.8 |
| Pick-a-Pic | ImageReward | 0.35 |

Clusters distribution of the datasets in Figure 4 and KL divergence between cluster distributions (Table 2) reveal Picsart Image-Social to exhibit the most pronounced dissimilarity from the others. Interestingly, the clusters where our training dataset manifests the highest density in comparison with ImageReward and Pick-a-Pic are characterized by themes such as background photos, flowers, interior and exterior design, and fashion, which are more aligned with popular content themes in visual creative art (see Figure 3).

### 3.3.2 Quantitative analysis

To gauge the effectiveness of existing solutions in predicting social popularity, specifically in the context of image editing, we conducted an evaluation of three models: PickScore (Kirstain et al., 2023), ImageReward (Xu et al., 2023a), and HPS v2 (Wu et al., 2023a), using the Picsart Image-Social test dataset utilizing their respective model evaluation code bases. Our primary evaluation metric was pairwise accuracy. Table 4 presents the results of this evaluation, with PickScore achieving the highest accuracy rate of 62.6%. Notably, this accuracy rate remains relatively modest.

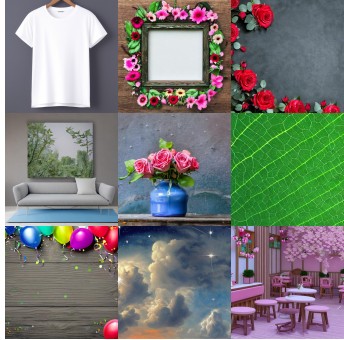

Figure 3: Generated images from prompt clusters distinct for creative editing

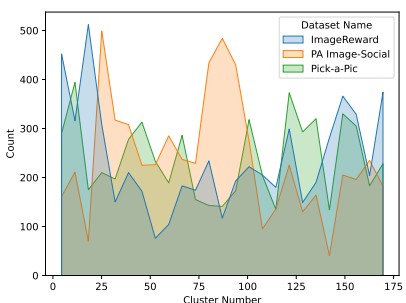

Figure 4: Cluster distribution of training prompts for different scoring models

This outcome can be primarily attributed to PickScore's data collection approach, which involved incorporating real user preferences, in contrast to ImageReward (60.48%) and HPS v2 (59.4%), which relied on annotations from moderators.

### 3.3.3 AGREEMENT OF CLIP AND LAION AESTHETIC SCORES WITH EXISTING METRICS

To draw insight from the relatively sub-optimal performance of existing score models on the Picsart Image-Social dataset we utilize vanilla CLIP (Radford et al., 2021) and LAION (Schuhmann et al., 2022) aesthetic models as estimators of text-image alignment and aesthetics/fidelity, since two of the three of analyzed models (HPS v2 and ImageReward) used those criteria as annotation guidelines during data collection stage.

By conducting the inference of vanilla CLIP and LAION aesthetic models on every score model's test sets (including Picsart Image-Social) and calculating pairwise accuracy, we can observe, as depicted in Table 3, that CLIP and LAION aesthetic scores exhibit the lowest levels of predictive efficacy on Picsart Image-Social dataset. This result underscores the notion that human preferences in our dataset reflect an image quality dimension that is discernibly separate from concepts of general fidelity and text-image alignment. This observation extends our understanding of the comparatively modest performance of other scoring models on the Picsart Image-Social dataset.

Table 3: Pairwise accuracy of LAION aesthetic and CLIP image-text alignment scores on test sets.

| Dataset | LAION aesthetic score | CLIP alignment score |
|---|---|---|
| Picsart Image-Social | 55.3% | 51.9% |
| Pick-a-Pic | 56.8% | 60.8% |
| ImageReward | 57.35% | 54.82% |
| HPD v2 | 72.6% | 62.5% |

## 4 SOCIAL REWARD: A NEW METRIC MODEL

In this section, we present our Social Reward model, which has been trained using user feedback collected from *Picsart*. We will describe the various backbone models and loss functions considered during our experiments, and conduct comparative analysis with existing solutions.

### 4.1 SOCIAL REWARD MODEL TRAINING

**Model:** Social Reward is trained through fine-tuning of the CLIP model. When provided with a prompt and an image, our scoring function calculates a real-valued score. This score is determined by representing the prompt using a transformer-based text encoder and the image using a transformer-based image encoder, both as $d$-dimensional vectors and then computing their cosine similarity. Fine-tuning the last two residual blocks of the textual encoder and the last three residual blocks of the visual encoder in CLIP yields superior results on our validation dataset.

**Loss function:** We experimented with various loss functions and observed that the triplet (Schroff et al., 2015) loss yielded the most favorable outcomes (please refer to Appendix B for details):

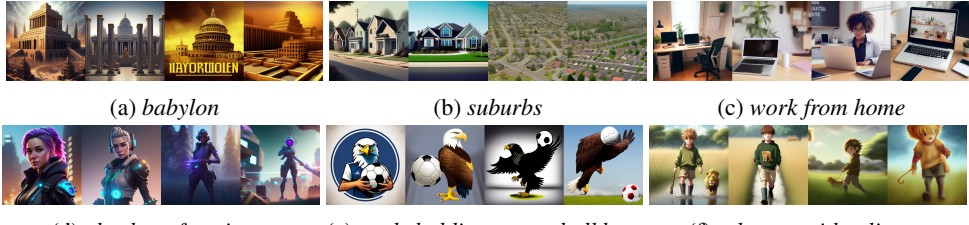

(a) *babylon*  (b) *suburbs*  (c) *work from home*

(d) *charlotte fortnite ...*  (e) *eagle holding soccer ball logo*  (f) *a boy ... with a lion ...*

Figure 6: Generated images ranked by Social Reward from best (*left*) to worst (*right*)

$$\mathcal{L}_{\text{triplet}}(\boldsymbol{a}, \boldsymbol{p}, \boldsymbol{n}) = \max(0, \|\boldsymbol{a} - \boldsymbol{p}\|^2 - \|\boldsymbol{a} - \boldsymbol{n}\|^2 + \alpha),$$

where $\boldsymbol{a}$ is the vector representation of the prompt, $\boldsymbol{p}$ is the vector representation of the positive image, $\boldsymbol{n}$ is the vector representation of the negative image and $\alpha$ is margin.

**Hyperparameters:** We employed the AdamW optimizer with a learning rate of 0.0003 and a batch size of 32, utilizing a distributed computing setup consisting of 8 A100 GPUs.

**Experiments:** We conducted a wide range of experiments including the exploration of various model backbones (such as CLIP and BLIP (Li et al., 2022)), fine-tuning with different components of these models (visual and textual encoders), and more ablation studies.

## 4.2 SOCIAL REWARD MODEL EVALUATION

We compared our model with existing solutions using a dedicated test dataset. We assessed our performance using the pairwise accuracy metric, based on prompt-image cosine similarity distance. As shown in Table 4 our model outperformed existing solutions in this evaluation. The codebase for Social Reward model training and evaluation is provided as supplementary material. Figure 6 demonstrates ranking of the images by Social Reward score (for more visuals see Appendix Figure 9).

To validate the results and check their generalizability we conducted a user study and compared with PickScore model, as it has the second highest accuracy on our dataset and also involves real users feedback. We generated 20 images with Stable Diffusion 2.1-base (Rombach et al., 2022) with the 100 prompts sampled from PickScore's test set and chose the best image by Social Reward and PickScore. Then we contacted number of popular creators from *Picsart* and collected their feedback with respect to which of the two images is more likely to get popular on the platform. Figure 5 shows Social Reward outperforms PickScore in ability to estimate social popularity and generalizes well to prompts outside of its training distribution.

Table 4: Models Comparison

| Model Name | Accuracy |
|---|---|
| Social Reward | **69.7%** |
| PickScore | 62.6% |
| ImageReward | 60.48% |
| HPS v2 | 59.4% |

Social Reward

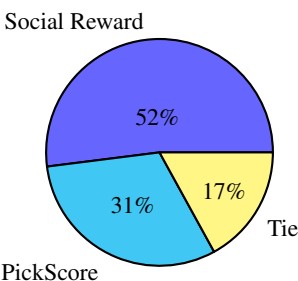

Figure 5: Human evaluation results using prompts from PickScore's test set.

## 4.3 USING SOCIAL REWARD TO FINE-TUNE T2I MODELS

**Training**: Social Reward can be used to fine-tune generative models to better align with the creative preferences of the community. For 1 million generated images by the users of our community, we calculated the Social Reward score and adapted the fine-tuning method from Wu et al. (2023b), for a given prompt choosing the image with the highest score and the image with the lowest score. For the image with the lowest score, we added a special token (*'t@y quality'*) to be used at inference time as a negative prompt. We also adapted the common practice of adding 625k subset of LAION-5B (Schuhmann et al., 2022) for regularization. The final training dataset consists of 330.000 images, half of which come from LAION's subset. We fine-tune both the U-Net and the textual encoder of Stable Diffusion 2.1-base model with a learning rate of 1e-5.

**Evaluation**: We calculate several metrics for images generated by vanilla and fine-tuned models. Evaluation is done on 2 separate sets of prompts: internal and general. Internal prompts are selected from our fine-tuning validation prompts set, while general prompts are taken from DiffusionDB (Wang et al., 2023). For each prompt, we generate 4 images with each model using DDIM scheduler (Song et al., 2022) for 30 steps. In Table 5, we present average scores and in Table 6 the percentage of times the fine-tuned model outperformed the baseline for each metric and prompt set. Visual comparisons are in Figure 7 and Appendix Figure 10.

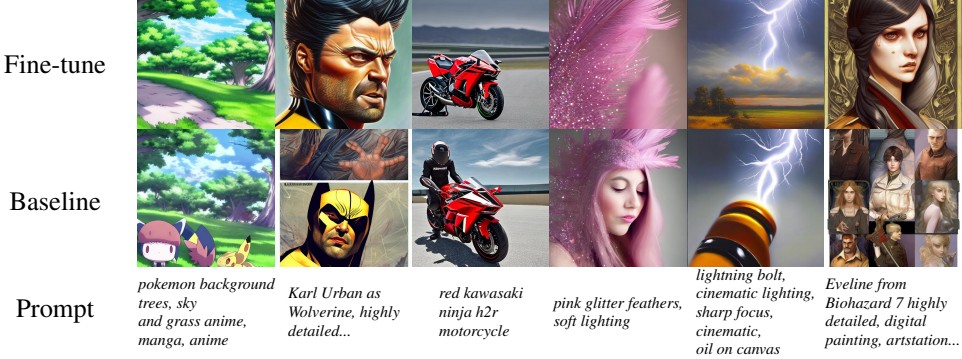

Figure 7: Generated image comparison between Baseline and Fine-tuned models

Table 5: Comparison between Baseline (SD-2.1-base) and fine-tuned models

| Model | Prompt | Social Reward | CLIP alignment | LAION aesthetic | HPS v2 | Image Reward | PickScore |
|---|---|---|---|---|---|---|---|
| Baseline | general | -0.095 | 0.280 | 5.902 | 0.259 | 0.117 | 0.199 |
| Fine-tune | general | -0.062 | 0.278 | 6.011 | 0.261 | 0.288 | 0.201 |
| Baseline | internal | -0.130 | 0.260 | 5.746 | 0.258 | -0.078 | 0.199 |
| Fine-tune | internal | -0.093 | 0.256 | 5.892 | 0.261 | 0.110 | 0.200 |

Table 6: Fine-tuned model win-rates vs Baseline

| Prompt | Social Reward | CLIP alignment | LAION aesthetic | HPS v2 | Image Reward | PickScore |
|---|---|---|---|---|---|---|
| internal | 75.6% | 43% | 73.8% | 73.4% | 70.6% | 63.2% |
| general | 75.6% | 43.8% | 71.2% | 71.4% | 70.6% | 67.8% |

CLIP alignment is the only metric with a slight decrease which can be explained by the fact that Picsart Image-Social's training data reflects social feedback of the image; users might compromise text-image alignment for the sake of selecting an image more suitable for creative editing. On the other hand, all other metrics have increased even on general prompts which shows that Social Reward not only assesses the potential image popularity in the context of creative editing but also incorporates to high extent notions of general aesthetic present in other scores.

## 5 CONCLUSION

In this work, we introduced a new concept known as Social Reward in the realm of text-conditioned image synthesis. Leveraging implicit feedback from social network users engaged with AI-generated images, we addressed the unique challenge of evaluating text-to-image models in the context of social network popularity and creative editing. Our comprehensive journey involved dataset curation from *Picsart*, resulting in the creation of the Picsart Image-Social dataset, a million-user-scale repository of implicit human preferences for user-generated visual art. Through rigorous analysis and experimentation, we demonstrated that our Social Reward model outperforms existing metrics in capturing community-level creative preferences. Moreover, we demonstrate that employing Social Reward to fine-tune text-to-image models leads to improved alignment not only with Social Reward, but also with other established metrics. By introducing this novel approach, we aim to enhance the co-creative process, align AI-generated images with the majority of users' creative objectives, and ultimately facilitate the creation of more popular visual arts in the digital space.

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

# A PICSART IMAGE-SOCIAL DATASET DETAILS

## A.1 HUMAN PREFERENCE DATATSETS' PROMPT ANALYSIS

To better illustrate differences between training sets of existing scoring models, we used t-SNE (van der Maaten & Hinton, 2008) method on the embeddings presented in 3.3.1 to map the prompts to 2 dimensions. Results in Figure 8 confirm Picsart Image-Social's distinctness from other sets.

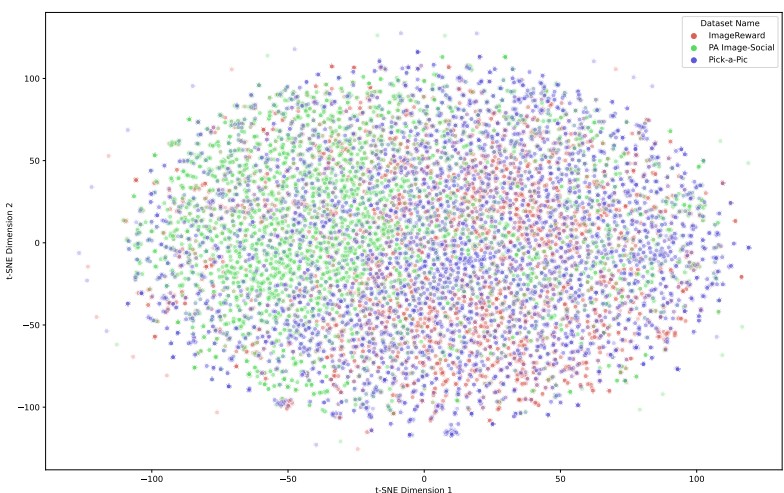

Figure 8: t-SNE visualization of training prompts for different scoring models

## A.2 DATA COLLECTION PROCEDURE

We initiate the data collection process by identifying the most popular synthetic images on *Picsart*, focusing on those with the highest number of remixes over a 10-month period. These images are then arranged based on remix count in descending order. The top 1% of these images are categorized as positive images, utilizing the "content signal" approach. Furthermore, if an image has been remixed by an influencer user, it is also included in the set of positive images, employing the "creator signal".

Subsequently, we gather prompts associated with the selected positive images. From these prompts, we retrieve all images that have garnered at least one remix and analyze the distribution of view counts before the first remix. By cutting this distribution at the 99th percentile, we establish a view count threshold for subsequent steps.

For negative images, we identify all images, linked to the same prompts, that have received zero remixes (there are large amounts of this kind of images due to "Pareto-like" nature of content diffusion in social platforms). For each prompt, we filter out "zero-remix" images falling below the established view count threshold. This process ensures that only images with sufficient evidence of negative user feedback are labeled as such.

Throughout the data collection procedure, all cases which cause inconsistencies are dropped. For instance, we omit prompts lacking images other than those labeled as positive or prompts with no images receiving zero remixes, etc.

To further refine the dataset, we employ in-house mature content detection models to filter out images classified within the NSFW category. This comprehensive approach enhances the reliability and accuracy of our collected data.

### A.3 DATASET IMPORTANT CHARACTERISTICS

Out of about 1.7 million images popular ones (deemed as positive) constitute 8%. The main reason behind the smaller number of positive images in Picsart Image-Social dataset is typical "Pareto like" pattern observed in majority social networks where minority of the content gathers majority of community engagement.

"Creator signal" that had been leveraged in collecting positive samples for Picsart Image-Social dataset is presented by 174 influencer users. Users are categorized as influential in *Picsart* if they have a certain amount of followers, are frequently posting new content, which gathers large volumes of community remixes.

Engagement levels of individual users, in our case, expectedly follow power law distribution logic (pattern which usually describes user activity in online social platforms). For instance, the least active 50% of our users generate about 30% of remixes. Whereas the most active 10% generate about 40% of remixes.

## B LOSS EXPERIMENTS

We investigated a range of loss functions, such as the Binary Cross Entropy, Reweighted Cross-Entropy Loss (Wang et al., 2021), and metric losses such as InfoNCE (van den Oord et al., 2019), Contrastive (Khosla et al., 2021) and Triplet (Schroff et al., 2015). The best results were obtained using the Triplet loss. Performance comparison of the model trained under different loss functions is presented in the Table 7.

Table 7: Different Losses and their Accuracies

| Loss | Accuracy |
|---|---|
| Triplet | 69.7% |
| InfoNCE | 68.5% |
| Contrastive | 67.1% |
| BCE | 67% |
| Reweighted Cross-Entropy | 65.9% |

## C ADDITIONAL VISUALIZATIONS

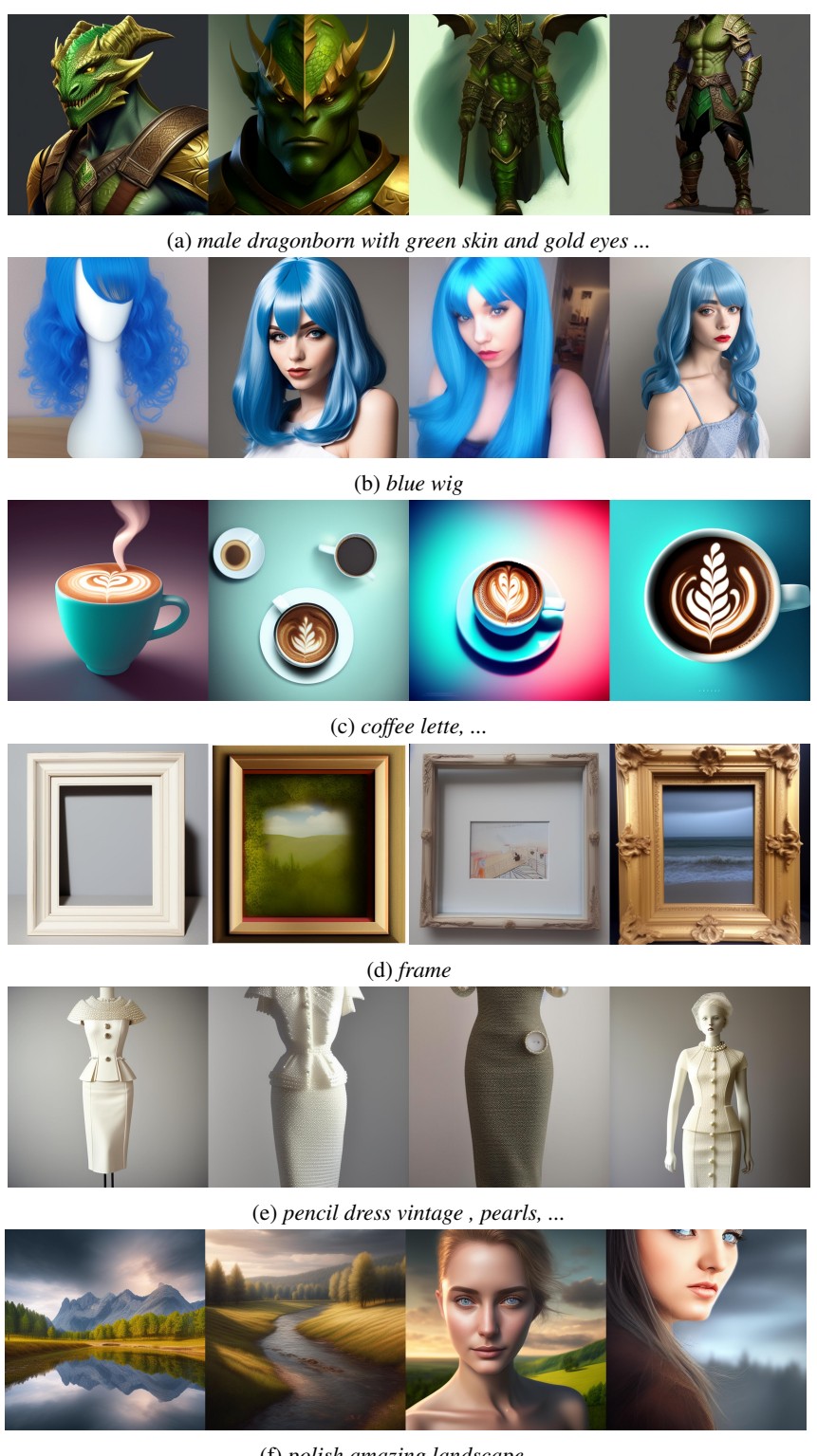

(a) *male dragonborn with green skin and gold eyes ...*

(b) *blue wig*

(c) *coffee lette, ...*

(d) *frame*

(e) *pencil dress vintage , pearls, ...*

(f) *polish amazing landscape, ...*

Figure 9: Generated images ranked by Social Reward from best *(left)* to worst *(right)*

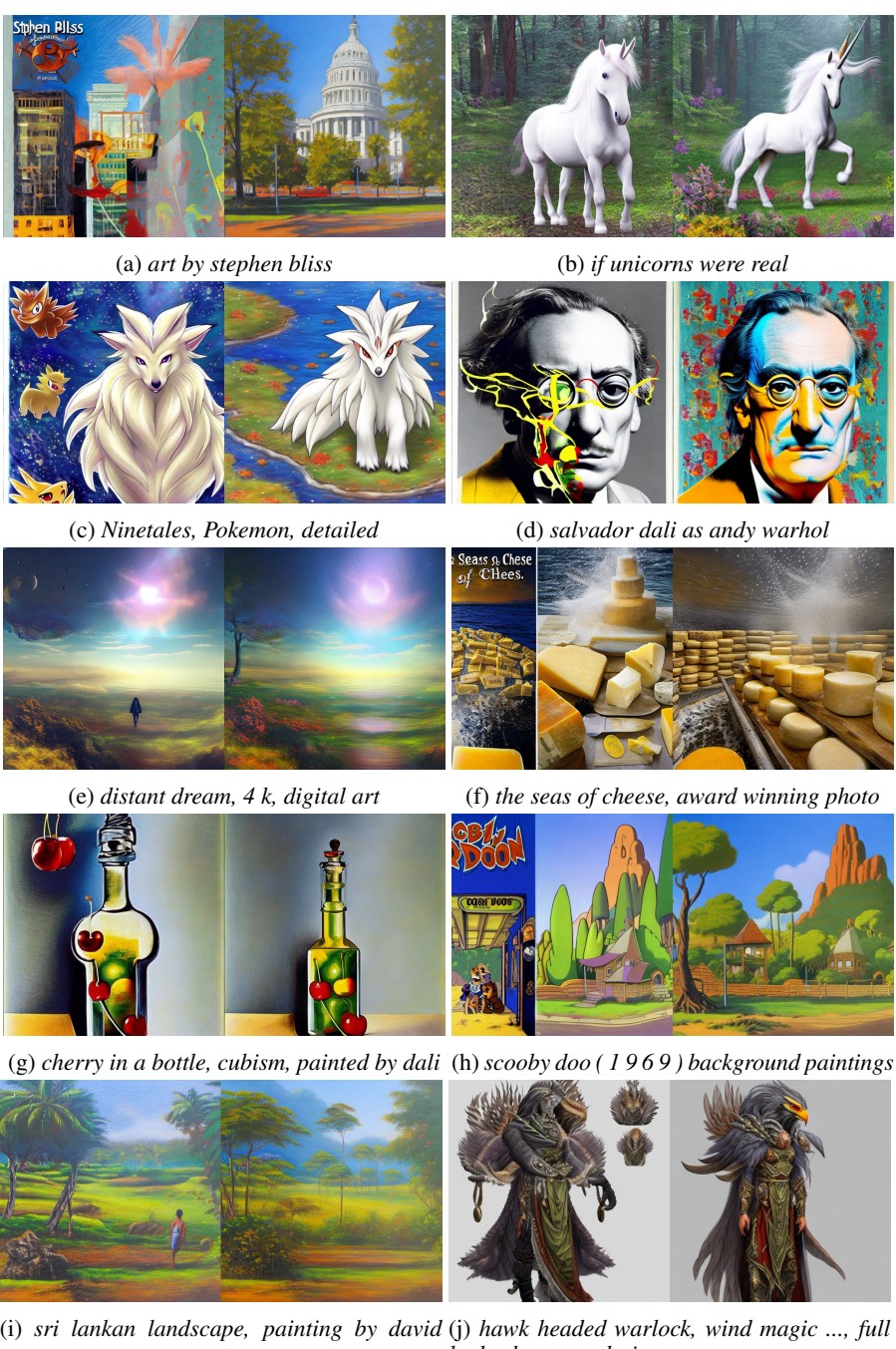

(a) *art by stephen bliss*

(b) *if unicorns were real*

(c) *Ninetales, Pokemon, detailed*

(d) *salvador dali as andy warhol*

(e) *distant dream, 4 k, digital art*

(f) *the seas of cheese, award winning photo*

(g) *cherry in a bottle, cubism, painted by dali*

(h) *scooby doo ( 1 9 6 9 ) background paintings*

(i) *sri lankan landscape, painting by david paynter,*

(j) *hawk headed warlock, wind magic ..., full body character design, ...*

Figure 10: Generated image comparison between Baseline *(left)* and Fine-tuned *(right)* models using prompts from DiffusionDB

