# OpenReview forum: "Social Reward: Evaluating and Enhancing Generative AI through Million-User Feedback from an Online Creative Community"
_ICLR.cc/2024/Conference — ICLR 2024 spotlight_

### Official Review · Reviewer_Poto · 2023-10-29

**Soundness:** 3 good
**Presentation:** 2 fair
**Contribution:** 2 fair
**Rating:** 5
**Confidence:** 4

**Summary:**

This paper proposes to create a new dataset of implicit human creative preferences for user-generated visual art. The dataset is created based on an online visual creation and editing platform and millions of user feedback from the online social community are collected to evaluate the quality of generated images. Due to the uniqueness of this dataset in modeling content creativity, the authors also proposed a new social reward metric to evaluate image quality. The social reward metric is further used to fine-turn the text-to-image generation model.

**Strengths:**

1.	Previous text-to-image generation human preference datasets mainly focus on general fidelity and text-image alignment, ignoring content creativity. This paper created a new dataset to fill this gap.
2.	To evaluate the creativity of the image, the authors utilize the number of times an image has been reused for editing purposes by other users, which is reasonable and more aligned with real users. The number of collected feedbacks is also in the million scale.
3.	The authors also demonstrate the proposed social reward metric's effectiveness in fine-turning the text-to-image generation model.

**Weaknesses:**

1.	In the proposed social reward metric, the authors mainly focus on the creation-related metric such as the number of times an image has been reused for editing purposes by other users. However, user comments, views or likes are also important dimensions. It’s better to show whether a higher value of the proposed metric in this paper will induce a lower or higher value of the traditional metrics.
2.	The social reward metric in this paper is optimized using the triplet loss. As a dataset and benchmark paper, it’s recommended to compare with other existing contrastive learning methods such as NCE or infoNCE.
3.	The constructed dataset consists of prompt, positive image, and negative image. It’s unclear what’s the specific threshold to decide the positive and negative image.

**Questions:**

See the Weaknesses for questions.

---

> ### Author Response · Authors · 2023-11-15
>
> Please accept our heartfelt appreciation for your diligent and insightful review of our manuscript. Your detailed observations and suggestions have shed light on several aspects of our paper that require further attention and refinement, and we believe they will significantly enhance the overall quality of the manuscript.
> With respect to the reasoning behind prioritizing remixes as user feedback metric (**weakness #1**) there are several considerations:
> - Since our platform revolves around creative editing and sharing images, users are naturally motivated to express themselves visually through remixes. This can be illustrated by remixes being made by our user base **25** times more than like and **40** times more than comment. Thus remix provides a much richer source of user feedback.
> - Remixes inherently involve a more active form of engagement compared to more passive actions like views, comments or likes. Users not only appreciate the image but actively participate in the creative process by remixing it, showcasing a deeper level of involvement and investment in the content thus adding a layer of quality to the engagement metric.
> - Directly using views as a metric may not accurately reflect user engagement because views only measure passive consumption, providing no indication of user interaction or appreciation. It doesn't distinguish between users who briefly glance at an image and those who actively engage with it.

---

> ### Author Response · Authors · 2023-11-15
>
> **The response to the weakness #2:**
> We agree with your suggestion of comparing the model performance trained with other contrastive learning based losses. In fact, we were running experiments with different losses and observed limited impact of loss on model accuracy.
> For instance, model trained with more conventional Contrastive loss produced 67.1 % in pairwise accuracy metric on test set. Suggested by you InfoNCE yielded a higher 68.5% on the test set. But since the best performance was achieved by Triplet loss (69.7%) it was the one included in the paper.
> Model performance comparison with different loss functions will be enclosed to the paper submission as an appendix.

---

> ### Author Response · Authors · 2023-11-15
>
> **The response to the weakness #3:**
> In order to address you question better please find below a detailed description of PA Image-Social dataset collection procedure. This description will also be added to the Appendix part of our manuscript.
> - We start from discovering the most popular synthetic images in Platform A in terms of accrued remixes within a 10 month time frame.
> All images are sorted by remix number in descending order.
> - Top 1% of the images are selected as positive images (in paper this approach is called “content signal”, section 3.2).
> Additionally, in case if the image had been remixed by an influencer user it is also added to the set of positive images (in paper this approach is called “creator signal”, section 3.2).
> - Then we retrieve prompts associated with collected positive images.
> - For the obtained prompts, we retrieve all the images that received at least one remix and construct the distribution of view count before the first remix.
> - We cut that distribution by 99% percentile and use the obtained view count threshold in the next steps.
> - For the same prompts, we retrieve all the images that received exactly zero remixes (there are large amounts of “zero remix” images due to “Pareto like” nature of content diffusion in social platforms).
> - For every prompt, we filter out “zero remix” images that are below view count threshold. In such a way, only images for which there is enough evidence of negative user feedback will be labeled as such.
> - During data collection procedure some cases which cause inconsistencies are dropped, for example, we drop prompts for which there are no images other than the ones labeled as popular or there are no images that received zero remix etc.
> - Additionally, we employ in-house mature content detection models to filter out images that fall within the NSFW category.

---

### Official Review · Reviewer_o2eb · 2023-10-30

**Soundness:** 3 good
**Presentation:** 3 good
**Contribution:** 3 good
**Rating:** 8
**Confidence:** 3

**Summary:**

This paper proposes a novel reward modeling framework, called Social Reward, for assessing community appreciation of AI-generated artworks. The framework leverages implicit feedback from social network users engaged in creative editing of generated images, rather than relying on limited size user studies guided by image quality and alignment with prompts.

The authors curate a million-user-scale dataset of implicit human preferences for user-generated visuals by drawing from an anonymous online visual creation and editing platform. Rigorous quantitative experiments and user studies show that the Social Reward model aligns better with social popularity than existing metrics.

**Strengths:**

The paper starts with an interesting insight. Social reward is a form of community recognition that provides a strong source of motivation for users of online platforms to actively engage and contribute with content to accumulate peers' approval. Positive social feedback are essential for maintaining social cohesion and individual well-being. On online social platforms, users seek satisfaction via accumulation of their network’s peers engagement with shared content in the form of likes, views, etc. Therefore, social reward can motivate users to engage with online platforms by providing them with a sense of validation and recognition from their peers.

The authors’ analysis exposes the shortcomings of current metrics in modeling community creative preference of text-to-image models’ outputs. Many quantitative experiments and user study show that Social Reward model aligns better with social popularity than existing metrics.

The dataset curation process for the million-user-scale dataset of implicit human preferences for user-generated visuals involved drawing from an anonymous online visual creation and editing platform. The dataset was curated by leveraging collective feedback, which implies multiple users’ editing choices for the given content item, as a cleaning mechanism of organic implicit user behavior. A number of other data collection techniques had been utilized for addressing such biases as caption bias, content exposure time, and user follower base biases. Each positive and negative instance within the PA Image-Social dataset is a product of collective, independent, implicit voting by user community. I think this dataset would be valuable for many follow-up analyses.

**Weaknesses:**

As one limitation, this paper does not aim to introduce a specific new RLHF algorithm as a primary focus. Instead, its main emphasis lies on exploring a novel aspect of data-centric reward modeling, which has previously received little attention. The paper introduces a comprehensive end-to-end solution, and it's important to acknowledge that each component of this solution builds upon prior ideas and engineering techniques. However, it's worth noting that the integration of these components into a cohesive whole represents a significant and commendable achievement.

**Questions:**

How was user data right and privacy protected in your data curation process? That was not mentioned anywhere in paper.

---

> ### Author Response · Authors · 2023-11-14
>
> We want to extend our gratitude for the time and effort you have invested in reviewing our work. Your in-depth analysis of our manuscript shows genuine interest in our research and serves as a great source of inspiration for future creative endeavors.
> - You correctly pointed out that the purpose of this work isn't the introduction of the novel RLHF framework, but establishing and exploring new dimension in human preference reward modeling.
> - With respect to the data privacy question Platform A is a company that complies with GDPR, CCPA, and other data protection legislation, and is collecting, storing, and processing users' data in accordance with the consent received. The Platform also allows the users to opt-out of certain processing purposes, as requested.

---

### Official Review · Reviewer_yu8R · 2023-10-30

**Soundness:** 4 excellent
**Presentation:** 3 good
**Contribution:** 4 excellent
**Rating:** 8
**Confidence:** 4

**Summary:**

This paper advocates the importance of social rewards as a motivating factor for user engagement and content contribution on online platforms. It focuses on the co-creative process in text-conditioned image synthesis within online social networks, emphasizing the challenges of assessing models in the context of collective community preferences. The paper introduces a novel framework called "Social Reward" that leverages implicit feedback from users engaged in creative editing of generated images. This approach is supported by an extensive dataset and demonstrates improved alignment with social popularity compared to existing metrics.

**Strengths:**

The main strengths and innovations of this paper are:

(1)	The introduction of Social Reward, a novel reward modeling framework that leverages implicit feedback from social network users engaged in creative editing of generated images, to assess community appreciation of AI-generated artworks. To put their contributions in context, the authors discussed recent endeavors that propose modeling human preference by learning scoring functions from datasets consisting of human-annotated prompts paired with synthetic images.

(2)	The curation of a million-user-scale dataset of implicit human preferences for user-generated visuals by drawing from an anonymous online visual creation and editing platform, perpetually invigorated by a vibrant user community, where the feedback is drawn from individuals who actively engage with the images on the platform for editing purposes. It is used to demonstrate the distinctiveness of the Social Reward in comparison to those employed by existing solutions and to highlight the limitations of these solutions.

(3)	The paper contributed to the field of T2I synthesis by introducing a new paradigm for evaluating the quality of AI-generated artworks that is more closely aligned with users' creative goals.  The rigorous quantitative experiments and user studies that show that the Social Reward model aligns better with social popularity than existing metrics. Social Reward can be used to fine-tune generative models to better align with creative preferences of the community.

Overall, this paper represents a nice contribution to the field of generative AI art and has the potential to improve the quality of AI-generated artworks by better aligning them with users' creative goals.

**Weaknesses:**

-	I failed to follow what the authors meant by referring to Pick-a-Pic as “relatively small scale of collected user preferences along with absence of ‘collective feedback’”. Please explain why.

-	In Figure 7, it is hard to tell why Social Reward fine-tuning actually improves, as the examples might just be cherry picked and observing them doesn’t give me the intuition what Social Reward essentially improves or corrects any pattern consistently. Table 6 numbers are also marginally close to each other. Could the authors elaborate some critical win cases or “take home points”, that can be consistently observed after using Social Reward fine-tuning?

-	The study relies entirely on self-report survey measures. Self-report can be subject to biases like social desirability bias where participants answer in a way they feel is more socially acceptable rather than reflecting their true thoughts and behaviors. How the authors think their data curation was/was not affected by the social bias?

-	(Minor) For robust user study, as a cross-sectional study collecting data at only one time point, it cannot determine causation or the direction of effects. A longitudinal design collecting multiple waves of data over time would allow for stronger conclusions about how the variables influence each other over time.

**Questions:**

Please check the weakness part. The first three points are more crucial for me.

---

> ### Author Response · Authors · 2023-11-14
>
> We would like to express our appreciation for your thoughtful review, which can be instrumental in improving the quality of our work. In response to your feedback, we would like to address the specific points you raised.
> **The response to the weakness #1:**
> Under the relatively small scale of collected user preferences in Pick-a-Pic dataset we mainly meant comparison with our PA Image-Social dataset. For instance, we have about 2.5 times more images (656K vs 1.7 M), about 5 times more image pairs (615 K vs 3M) and about 2.5 more prompts (38.5 K vs 104 K). Please refer to Table 1. But most importantly is 230 times difference in the number of users involved (6.4 K vs 1.5 M). This number is critical in the light of our research goal, evaluating the performance of  text-to-image models within the context of social network popularity for creative editing, because of two reasons:
> - It is necessary for the collected human preference to be from a large group of users to be deemed representative of community scale feedback
> - The authors of  Pick-a-Pick built a web application, where users can generate images and express their preferences on them. Each prompt-image pair is getting feedback from a single user (the one who generated the image) in contrast to **”collective feedback”** (which implies multiple users’ independent feedback on the same image), that is the principle upon which PA Image-Social dataset is built.

---

> ### Author Response · Authors · 2023-11-14
>
> **The response to the weakness #2:**
> For more visuals of generated image comparison between baseline and fine-tuned models please refer to **figure 3** of supplementary material appendix file named **“SocialReward_Appendix_ICLR_2024_Conference_Submissions.pdf”**.
> We believe numbers reported in **Table 6** showcase significant performance gain of fine tuned model over the baseline. The percentage values in the table signify the percentage of times an image generated by a fine-tuned model is favored in terms of specific metric over the one generated by baseline model for the given prompt. We can observe that with exception of CLIP alignment, all scores are favoring fine tuned versions in at least 63% of cases and up to 75% of cases.
> With respect to the consistent patterns observed  in text-to-image models fine-tuned using SocialReward here are some of them:
>
> 1. **Improved quality:**
>  - Overall quality/fidelity of generated images visibly improved which can be noticed in decreased number of artifacts, distorted objects and shapes, more minimalistic and higher aesthetic quality of generated images
>
> 2. **Complete Display of Main Objects:**
>   - The model consistently generates images where the main object is fully displayed, minimizing instances of partial or cropped representations. This characteristic ensures that the central focus of the image is readily available for creative manipulation.
>
> 3. **Preference for Minimal Text:**
>   - There is a notable tendency for the model to avoid generating unnecessary text in the images, emphasizing a visual-first approach and compensating for the weakness of current models to generate text without artifacts.
>
> 4. **Reduced Collages and Merged Images:**
>   - There is a decrease in the generation of complex image collages or the merging of multiple images. This design choice provides users with a clean canvas, facilitating easier integration of additional elements during the creative editing phase.
>
> 5. **Clarity in Backgrounds, Especially in Natural Scenes:**
>   - The model consistently produces images with clear and distinct backgrounds, particularly noticeable in natural scenery. The absence of additional objects in the background enhances the usability of the generated images for creative projects, offering a blank slate for further customization.

---

> ### Author Response · Authors · 2023-11-14
>
> **The response to the weakness #3:**
> While social bias presents an important concern in terms of collecting candid and reliable responses in self-report surveys, we believe its impact within the scope of our study is limited.  We leverage organic user behavior as a source of our data. Users are not asked to participate in the survey, solely implicit anonymous feedback in the form of users organically choosing images for editing purposes is used.
> However, we fully share the concern of  accounting for biases in data collection procedure from such noisy source as social network and specifically address such biases as Content Exposure Time Bias, User Follower Base Bias and Prompt Bias **(section 3.2 Data Collection).**

---

> ### Author Response · Authors · 2023-11-14
>
> **The response to the weakness #4:**
> Thank you for your very valid remark regarding the longitudinal design of data collection.
> Although our study may not establish causation or the direction of effects in the same way a classic
> longitudinal design would, the extended data collection period can still provide valuable insights into how
> variables may influence each other over time. Due to this consideration of the importance of having a representative dataset in temporal dimension, in the scope of this work we collected data from a prolonged period of 10 months in the scope.
> We plan to dive into a longitudinal study for user preference shift and how that might lead to our score model updating over time, as the scope of our next work.

---

> > ### Comment · Reviewer_yu8R · 2023-12-03
> > **Most of the concerns are resolved.**
> >
> > The authors have resolved most of my concerns. I will raise the score.

---

### Official Review · Reviewer_hCCD · 2023-10-30

**Soundness:** 3 good
**Presentation:** 3 good
**Contribution:** 3 good
**Rating:** 6
**Confidence:** 2

**Summary:**

This paper presents a groundbreaking approach by introducing the concept of utilizing social rewards within a reward modeling framework to assess and enhance generative AI systems. The authors collected data from Platform A and employed the frequency with which an image is reused for editing by other users as a performance metric. Notably, they have constructed a colossal dataset, known as the "PA Image-Social dataset", which collects feedback on the relationships between images and prompts, offering a valuable resource for the analysis of positive and unpopular image-prompts associations.

The study also includes a comprehensive evaluation of the Social Reward model, demonstrating its superiority over existing models in capturing community-level creative preferences. Furthermore, the authors conducted an evaluation showcasing the effectiveness of fine-tuning AI models with Social Reward, revealing performance improvements across multiple performance metrics. This innovative paper brings forth a promising avenue for the assessment and enhancement of generative AI, backed by meticulous research and empirical evidence.

**Strengths:**

Thanks for your interests in ICLR! Overall, this is an interesting paper on a topic which is of interest to ICLR Conference. It introduces an innovative set of metrics designed to gauge the intricate interplay between prompts and images, accompanied by the construction of an unprecedented million-user-scale dataset, offering invaluable insights into human preferences for user-generated visual art. The metrics, composed of five distinct factors, provide a robust framework for evaluating performance.

Additionally, the authors furnish a robust evaluation of the Social Reward model, including comprehensive comparisons with existing models and metrics. Their meticulous analysis strengthens the paper's standing in the field and underscores its potential to advance our understanding of generative AI.

**Weaknesses:**

The paper offers a comprehensive introduction to the metrics and evaluation methods; however, I am very curious about the PA Image-Social dataset and want to learn more about its details.

**Questions:**

I am curious about the data distribution within the collected dataset. It would be highly informative if the authors could provide insights into key aspects such as the percentage of popular images, the presence of influential users, and the overall engagement levels of individual users.

---

> ### Author Response · Authors · 2023-11-14
>
> We would like to express our sincere gratitude for your thoughtful and constructive review of our paper submission.
> We appreciate the time and effort you have invested in reviewing our work.
> With respect to the question you raised, firstly, let us note that our paper indeed can greatly benefit from incorporating more information regarding PA Image-Social dataset. On the second note, here are the details you requested:
> - Popular (deemed as "positive") images constitute 8% of all images in PA Image-Social dataset. The main reason behind the smaller number of “positive” images in PA Image-Social dataset is typical “Pareto like” pattern observed in majority social networks where minority of the content gathers majority of community engagement.
> - There are 174 influencer users (users are categorized as influential in our platform if they have a certain amount of followers and are frequently posting new content) whose remixing of generated images contributed to the dataset collection.
> - Engagement levels of individual users, in our case, expectedly follow power law distribution logic (pattern which usually describes user activity in online social platforms). For instance, the least active 50% of our users generate about 30% of remixes. Whereas the most active 10% generate about 40% of remixes.
> - We plan to include this information in the paper’s Appendix section.

---

### Author Response · Authors · 2023-11-20

Thank you once again all reviewers for elaborate and insightful feedback!
We have incorporated the information from our responses into manuscript **(Appendix section A2, section A3, section B)** along with some minor formatting/typos related fixes in the manuscript file.
Additionally, the **appendix has been removed from supplementary material and has been added to the main manuscript**.
Now supplementary materials contain **only code.**
Thank you!

---

### Meta-Review · Area_Chair_JYZm · 2023-12-06

**Metareview:**

The authors introduce the concept of social reward as a method for improving text to image generation within online communities.  This paradigm shift in reward modeling leverages collective implicit feedback from social network users who specifically employ generated images for creative purposes. This distinction underscores the relevance and applicability of Social Reward to the creative editing process, providing a more faithful estimation of alignment between AI-generated images and community-level creative preference.  The authors highlight that Existing evaluation methods predominantly center on limited size user studies guided by image quality and alignment with prompts, in contrast this paper seeks to optimize a far nosier signal of “image popularity.”  In addition to evaluation, the authors outline their journey for creating this dataset and collecting feedback.

The authors provide a rigorous quantitative experiments and user study show that their proposed “Social Reward” model aligns better with social popularity than existing metrics. They utilize Social Reward to fine-tune text-to-image models, and show that the images generated from their fine-tuned model are more favored by also other established metrics.

Strengths:
-The introduction of Social Reward, a novel reward modeling framework that leverages implicit feedback from social network users engaged in creative editing of generated images,
-Previous text-to-image generation human preference datasets mainly focus on general fidelity and text-image alignment, ignoring content creativity. This paper created a new dataset to fill this gap.
-The curation of a million-user-scale dataset of implicit human preferences for user-generated visuals by drawing from an anonymous online visual creation and editing platform.
-The paper contributed to the field of T2I synthesis by introducing a new paradigm for evaluating the quality of AI-generated artworks that is more closely aligned with users' creative goals
-The authors also demonstrate the proposed social reward metric's effectiveness in fine-turning the text-to-image generation model.

Weaknesses:
- one reviewer feels that the authors should have considered established social metrics such as "likes," "comments" and "shares" as part of thier social reward metric.
-The social reward metric in this paper is optimized using the triplet loss. As a dataset and benchmark paper, it’s recommended to compare with other existing contrastive learning methods such as NCE or infoNCE.
-. It’s unclear what’s the specific threshold to decide the positive and negative image.
-it is hard to tell why Social Reward fine-tuning actually improves, as the examples might just be cherry picked and observing them doesn’t give me the intuition what Social Reward essentially improves or corrects any pattern consistently.
-The study relies entirely on self-report survey measures. Self-report can be subject to biases like social desirability bias where participants answer in a way they feel is more socially acceptable rather than reflecting their true thoughts and behaviors. How the authors think their data curation was/was not affected by the social bias?
-each component of this solution builds upon prior ideas and engineering technique (this was listed under "weaknesses" but the overall tone of the comment is quite positive)

**Justification For Why Not Higher Score:**

The paper has some detractions, but not many.

**Justification For Why Not Lower Score:**

Overall, given the interest in the field in collecting human feedback (for various metrics) I feel that this paper might be of widespread interest to the ICLR audience and might do well as a spotlight talk.

---

### Decision · Program_Chairs · 2024-01-16

Accept (spotlight)